# Anisotropic growth is achieved through the additive mechanical effect of material anisotropy and elastic asymmetry

**Firas Bou Daher[1,2], Yuanjie Chen[2], Behruz Bozorg[2,3], Jack Clough[2], Henrik Jönsson[2,3,4], Siobhan A Braybrook[1,2,5]***

[1]Department of Molecular, Cell and Developmental Biology, University of California, Los Angeles, Los Angeles, United States; [2]The Sainsbury Laboratory, University of Cambridge, Cambridge, United Kingdom; [3]Computational Biology and Biological Physics Group, Lund University, Lund, Sweden; [4]Department of Applied Mathematics and Theoretical Physics, University of Cambridge, Cambridge, United Kingdom; [5]Molecular Biology Institute, University of California, Los Angeles, Los Angeles, United States

*For correspondence:
siobhanb@ucla.edu

**Competing interests:** The authors declare that no competing interests exist.

**Abstract** Fast directional growth is a necessity for the young seedling; after germination, it needs to quickly penetrate the soil to begin its autotrophic life. In most dicot plants, this rapid escape is due to the anisotropic elongation of the hypocotyl, the columnar organ between the root and the shoot meristems. Anisotropic growth is common in plant organs and is canonically attributed to cell wall anisotropy produced by oriented cellulose fibers. Recently, a mechanism based on asymmetric pectin-based cell wall elasticity has been proposed. Here we present a harmonizing model for anisotropic growth control in the dark-grown *Arabidopsis thaliana* hypocotyl: basic anisotropic information is provided by cellulose orientation) and additive anisotropic information is provided by pectin-based elastic asymmetry in the epidermis. We quantitatively show that hypocotyl elongation is anisotropic starting at germination. We present experimental evidence for pectin biochemical differences and wall mechanics providing important growth regulation in the hypocotyl. Lastly, our in silico modelling experiments indicate an additive collaboration between pectin biochemistry and cellulose orientation in promoting anisotropic growth.

DOI: https://doi.org/10.7554/eLife.38161.001

## Introduction

In order for most dicot seedlings to emerge from the soil successfully, the hypocotyl must grow rapidly and anisotropically (*Baskin and Jensen, 2013*). Such tissue anisotropy is exhibited in many plant organs when directionality is key: roots moving through the soil, stems reaching upwards and climbing tendrils (*Baskin, 2005*). Anisotropy, in terms of differential growth, is defined as the relative change in principal dimensions over time; for example, in the young hypocotyl, there is an increase in length versus width. In the *Arabidopsis* hypocotyl, the direction of anisotropy (upwards) is relatively fixed but the magnitude of growth anisotropy (how fast) is presumed to change over time (*Gendreau et al., 1997*). This presumption is based upon measurements of cell length over time which indicate that a 'wave' of elongation runs acropetally from the base of the organ towards the cotyledons (*Gendreau et al., 1997*).

Plant cells are contained within a stiff cell wall thus the cell wall must change to allow growth of cells and, ultimately, organs (*Braybrook and Jönsson, 2016*). With respect to cellular anisotropy, growth may be generated by a cell wall which yields to (or resists) forces in a spatially differential

**eLife digest** Unlike animal cells, plant cells are surrounded by a stiff shell called the cell wall. Cell walls are composed of two main types of material: cellulose, the strong fibers that make up paper, and a pectin gel, which holds everything together.

In order for plants to grow, the cell wall has to yield to the pressure inside the cell and allow stretching. The direction of individual cell growth in plants is thought to be controlled by the direction of cellulose fibers in the wall; if they wrap around the cell like hoops on a barrel, the cell can only grow 'up' and not 'out'. Cellulose direction is dictated by the orientation of tracks inside the cell called microtubules. Another recent idea says that the pectin gel can control growth direction; if the side walls of a cell have less gelling they can elongate more, increasing upward growth. What had not been examined is whether cellulose and pectin might both contribute to directional growth.

Young seedlings emerge from the soil through the directional growth of the young stem, or hypocotyl. Using advanced microscopy, nano-materials testing, genetics techniques and computational models Bou Daher et al. studied the hypocotyl of a commonly studied plant called *Arabidopsis thaliana*. The results demonstrate that not only do both components of the cell wall control growth, but they work together from different tissues within the plant. The orientation of microtubules (and hence cellulose fibers) in cells in the inner tissues of the hypocotyl combines with pectin gelling in the outer tissue layer to produce fast, directional growth.

Understanding how directional growth is achieved could enable us to change it in useful ways. This could lead to a number of agricultural improvements. For example, many seedlings are lost as they first grow through the soil to reach the light, so improving directional growth could increase crop yields. In order to do this, researchers would need to explore how common the co-operative mechanism Bou Daher et al. have discovered is in other plant species (such as soybean, corn and wheat) and in other plant organs (like the adult stem and the roots).

DOI: https://doi.org/10.7554/eLife.38161.002

manner (*Baskin, 2005*). The cell wall is a complex material with a fibrillar cellulosic backbone within a pectin-rich matrix (*Cosgrove, 2016*). In the alga *Nitella*, cell wall structure has been proposed to regulate anisotropy through the coordinated orientation of cellulose fibers within the wall: circumferential wrapping of cellulose fibers restricts transverse growth and the passive (or active) separation of fibers allows axial growth leading to anisotropy (*Green, 1960*; *Probine and Preston, 1962*). This organization within a material, gives rise to directionally differential yielding to force and would make the wall material an anisotropic material. Material anisotropy can be tested by applying external force sequentially along two perpendicular directions and measuring the difference in yield: an anisotropic material would yield differently in the two directions. Consistent with this concept cellulose fiber orientation has been correlated with material anisotropy in *Nitella* (*Probine and Preston, 1962*) and in epidermal cells of onion and *Kalanchoe* leaves (*Kerstens et al., 2001*).

It is attractive to imagine that every cell within an anisotropically growing organ would display cellulose orientation perpendicular to growth, like *Nitella*. Indeed, this has been demonstrated in maize and *Arabidopsis* roots, the wheat leaf epidermis, rice coleoptiles, soybean hypocotyls and onion scales (*Baskin et al., 1999*; *Paolillo, 1995*, *Paolillo, 2000*; *Verbelen and Kerstens, 2000*; *Pietra et al., 2013*). However, there are many exceptions where the net cellulose orientation in the outer wall of the epidermis of elongating cells was not perpendicular to the axis of growth. These include rice and oat coleoptiles, *Arabidopsis* hypocotyls and roots, pea epicotyls and dandelion peduncles (*Paolillo, 2000*; *Verbelen and Kerstens, 2000*; *Iwata and Hogetsu, 1989*; *Roelofsen, 1966*). Cortical microtubule orientation may act as a proxy for newly-deposited cellulose orientation as in most cases they correlate strongly. Although some exceptions exist in root cells (*Himmelspach et al., 2003*; *Sugimoto, 2003*), the correlation has been very well documented in the case of *Arabidopsis* hypocotyls where microtubules, cellulose-synthase complex movement and cellulose microfibrils orientation are correlated in epidermal cells (*Paredez et al., 2006*). Most recently, transversely aligned microtubule orientation was observed in *Arabidopsis* hypocotyls on the inward

facing epidermal cell walls and those of inner cortical tissues, while the outer face of the epidermis presented as unaligned (*Crowell et al., 2011*; *Peaucelle et al., 2015*).

These data do not necessarily negate the hypothesis from *Nitella*, but instead underline the possibility that different cells in different tissues contribute to anisotropy differently. In complex multi-cellular organs like the hypocotyl it may not be necessary for each individual cell to mimic *Nitella*. Cortex cells exhibiting transverse cellulose or microtubule orientation could provide anisotropy to the epidermis through their physical connection. This sharing of information is consistent with the epidermal growth theory (for examples and reviews see [*Baskin and Jensen, 2013*; *Kutschera, 1992*, *Kutschera, 2008*; *Kutschera and Niklas, 2007*]): in growing plant organs, internal tissues can provide the force for growth but the act of growth only occurs once the epidermis, holding the tension, releases (*Baskin and Jensen, 2013*). To our knowledge, transverse microtubule or cellulose orientation in inner tissues alone has not been experimentally perturbed and so this hypothesis remains unconfirmed.

While it is often assumed that cellulose orientation *alone* confers anisotropy, experimental evidence points to further complexity. Disruption of cellulose orientation has mixed effects on cell-shape anisotropy: treatment with cellulose synthesis inhibitors reduces cell anisotropy in roots and hypocotyls (*Desprez et al., 2002*; *Heim et al., 1991*) with a developmentally stage-specific magnitude (*Refrégier et al., 2004*); the mutant *botero/katanin* has defects in microtubule orientation and shows reduced cell length but maintains some anisotropy (*Bichet et al., 2001*); mutations in cellulose synthase complex subunits cause a decrease in cell and organ length, but again some anisotropy is maintained (*Refrégier et al., 2004*; *Chen et al., 2003*; *Fagard, 2000*; *Fujita et al., 2013*); in some mutants early growth is normal when compared to wild-type (*prc1-1* [*Refrégier et al., 2004*]). These subtleties strongly indicate that there may be more to tissue anisotropy than cellulose orientation alone (*Baskin, 2005*). The pectin matrix of the cell wall arises as a strong candidate for regulating anisotropic growth as the transition from slow to rapid growth has been hypothesized to involve changes in pectin chemistry (*Pelletier et al., 2010*). It has recently been proposed that differential pectin rigidity, within individual epidermal cells, might dictate the onset of anisotropy (in the absence of epidermal MT orientation) (*Peaucelle et al., 2015*). Still, a quantitative understanding of the contribution of wall anisotropy via cellulose fibers and wall heterogeneity via pectin biochemistry is lacking. Here, we use an interdisciplinary approach to address the question: how is anisotropic growth in the hypocotyl achieved through cell wall mechanics?

## Results

### Hypocotyl epidermal cells exhibit anisotropic growth from germination

Historically, analysis of cell-level growth in dark-grown *Arabidopsis* hypocotyls focused on cell length alone (*Gendreau et al., 1997*) yet cell width over time is an important parameter for analysis of anisotropy. In order to deepen our understanding of anisotropic growth in the dark-grown hypocotyl, we undertook a detailed analysis of cell length and width from 0 hr post-germination (HPG) to 72HPG. *Arabidopsis* hypocotyls expressing a plasma-membrane YFP-tagged marker (*Willis et al., 2016*) were synchronized by selecting seeds at germination (T0; radical emergence from endosperm). At 6 hr intervals, 20 seedlings were sampled for confocal imaging, with a new set being imaged at each timepoint since confocal imaging stopped dark-grown hypocotyl elongation. In order to focus on cell growth, analyses were restricted to epidermal files with no division; we observed that some files underwent transverse anticlinal divisions during the first 24HPG (*Figure 1—figure supplement 1*; files lacking GL2 expression). Since there were no divisions in GL2-expressing files, cell indices were assigned by position along the hypocotyl. Non-dividing files had 17–19 cells starting from the collet (the transition zone between the root and the hypocotyl) and ending at the cotyledons base (*Figure 1A*). Utilizing this system, cell geometry could be analyzed in time and by cell position, to approximate cell-level growth dynamics.

Our analyses revealed that while cell length increased in an acropetal wave consistent with the literature (*Gendreau et al., 1997*; *Peaucelle et al., 2015*; *Refrégier et al., 2004*; *Pelletier et al., 2010*), cell width increased more slowly and evenly along the hypocotyl length (*Figure 1B*, *Figure 1—figure supplement 1*). These observations were consistent with differential regulation of axial and radial cell expansion. Also, in our system all hypocotyl cells were geometrically anisotropic

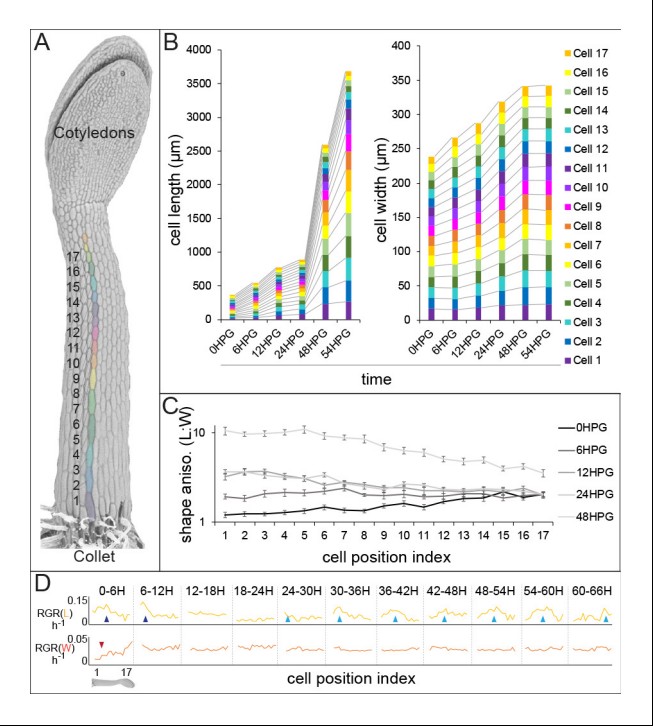

**Figure 1.** Hypocotyl epidermal cells exhibit a wave of growth in length and not in width. (**A**) A scanning electron micrograph of a 24HPG etiolated hypocotyl showing cell indices in a non-dividing cell file, numbered from the collet to the cotyledons. (**B**) Cell length and width by cell position index displayed from 0-54HPG. (**C**) Cell shape anisotropy (length:width) by cell index from 0-48HPG. (**D**) Relative growth rates (RGR) for length and width of cells by index from 0-66HPG, in 6H windows. Data in B-D were collected from 20 hypocotyls per time point imaged by confocal microscopy, from at least two non-dividing files per hypocotyl. The data in B are presented without error bars to make visualization possible; standard errors can be found in *Figure 1—figure supplement 1*. RGR values and standard errors can be found in *Supplementary file 1*. Blue arrowheads point to early-pulse growth adjacent to the collet, red arrowheads to early growth suppression, light blue arrowheads trace the maximal RGR(L) and proxy the acropetal wave.

DOI: https://doi.org/10.7554/eLife.38161.003

The following figure supplement is available for figure 1:

**Figure supplement 1.** Characterization of dividing and non-dividing cell files and growth in non-dividing files.
DOI: https://doi.org/10.7554/eLife.38161.004

from the time of germination, irrespective of position along the hypocotyl (*Figure 1C*; shape anisotropy, ratio of cell length to width). Calculations of relative growth rates for each cell index, over 6H intervals, revealed that cell length was always increasing at a higher rate than cell width (*Figure 1D* and Supplemental file 1; RGR(L) vs RGR(W)). Relative growth rate for cell length (RGR(L)) by position clearly demonstrated the movement of the acropetal wave, beginning around 24HPG (*Figure 1D*, light blue arrows). Interestingly, at very early time intervals (0-12HPG) there was a higher RGR(L) and a suppression of RGR(W) in basal hypocotyl cells, potentially as a holdover from germination (*Bassel et al., 2014*) (*Figure 1D*; blue and red arrowheads respectively). The RGR(L) of cells within the acropetal wave was relatively constant (8.95% ± 0.56 per hour; *Figure 1D*, *Supplementary file 1*), indicating that there was a transition from slow to rapid elongation within the wave but that growth rate did not increase over time as the wave moved. In contrast, the RGR(W) was, after the initial suppression, remarkably constant in time and space (2.6% ± 0.2 per hour; *Figure 1D*, Supplemental file 1). Our data paint an accurate picture of dark-grown hypocotyl growth: cells are geometrically anisotropic at germination, their growth is always anisotropic, and the acropetal wave is only evident in the elongation of cells but not their expansion in width.

# Anisotropically expanding hypocotyl epidermal cells do not exhibit strong transverse cortical microtubule orientation

Transverse cellulose orientation, or its proxy microtubule orientation, is commonly invoked to explain the mechanism of anisotropy. As hypocotyl cells displayed growth anisotropy from the time of germination, we examined whether they also exhibited transverse microtubule orientation. Microtubules were visualized by confocal microscopy imaging of hypocotyl basal epidermal cells. Imaging of *35S:: GFP-MAP4* (*Marc et al., 1998*) in dark-grown hypocotyls was conducted in short periods after exposure to light to prevent light induced reorientation, MT rearrangements or rotary movements as previously reported (*Chan et al., 2007*; *Lindeboom et al., 2013*; *Sambade et al., 2012*). MT images were recorded at 0HPG, 24HPG and 65HPG; representing the time of germination, the transition to rapid growth and the phase of rapid growth, respectively. MT angle was determined using MicroFilament Analyzer (*Jacques et al., 2013*). Both the inner and outer epidermal faces of hypocotyl cells were imaged, when possible, as they have been shown to exhibit different patterns of MT angles (*Crowell et al., 2011*).

At germination, basal epidermal cells exhibited a wide range of MT angles on their outer epidermal face with a slight transverse tendency (~30% presenting transversely between 0 and 10°; *Figure 2A,B*). We were unable to image deeper at this stage, likely due to the dense cell contents scattering the excitation and emission light. By 24HPG, the average angle on the outer face was slightly axial with ~31% of MTs being oriented between 80–90° (*Figure 2A,B*). These data correlate well with those recently reported for cells below the cotyledons before elongation in older

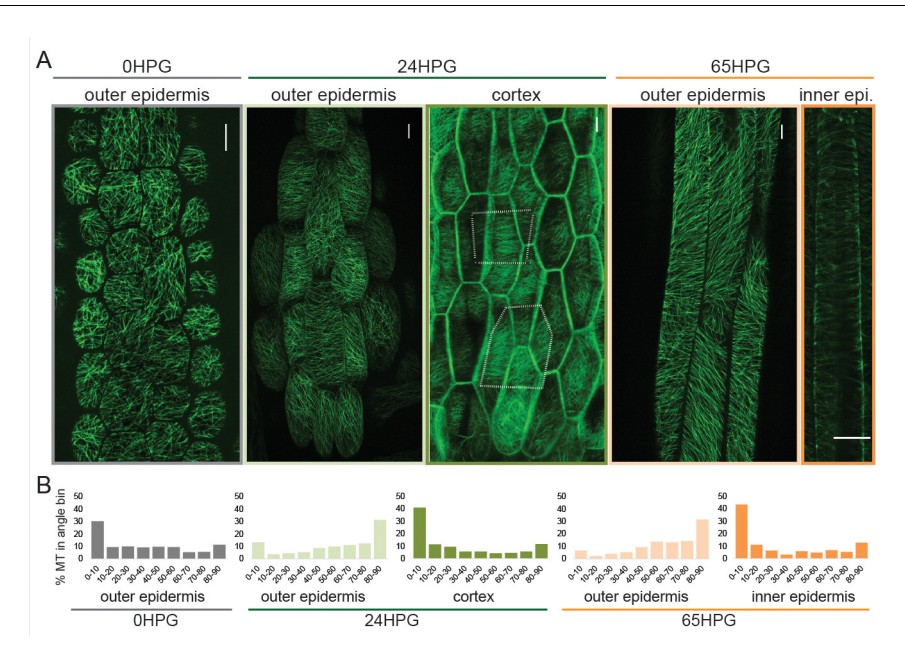

**Figure 2.** Microtubule alignment is weakly transversely aligned early in hypocotyl growth. (**A**) Representative images of microtubule organization as visualized with 35S::GFP-MAP4 at 0HPG, 24HPG and 65HPG. Scale bars = 10 µm. Location of images reported as: outer or inner epidermal face, or cortex. Dotted outlines in cortex image indicate cortex cell outlines. (**B**) Frequency distribution of microtubule angle grouped in 10 degree intervals from 0HPG to 65HPG using MicroFilament Analyzer (MFA); sample numbers were: 0HPG: n = 65 cells (from 5 hypocotyls); 24HPG: n = 30 (from 9 hypocotyls); 65HPG: n = 13 (from 6 hypocotyls). For 24HPG cortex analysis, n = 36 (from 5 hypocotyls). Examples of MTs at 24HPG outer epidermal faces visualized with *35S::GFP-TUA6, 35S:: GFP-EB1* and *CESA3::CESA3-GFP* are found in *Figure 2—figure supplement 1*.

DOI: https://doi.org/10.7554/eLife.38161.005

The following figure supplement is available for figure 2:

**Figure supplement 1.** A wider selection of MT markers and CESA3 used for verification.

DOI: https://doi.org/10.7554/eLife.38161.006

hypocotyls (*Crowell et al., 2011*). Similar patterns at 24HPG were observed with two other microtubule markers (GFP-TUA6 and GFP-EB1) and GFP-CESA3 (*Figure 2—figure supplement 1*; [*Chan et al., 2003*; *Desprez et al., 2007*; *Mathur et al., 2003*; *Ueda et al., 1999*]). At 24HPG, we could not observe MT signal at the inner epidermal face, but could at the adjacent cortex cell faces (a phenomenon consistent across all three marker lines, at this early stage). MTs at cortex cell faces appeared more transversely aligned and presented ~41% between 0 and 10° (*Figure 2A,B*). From these data we concluded that MTs at the outer epidermal face were weakly transverse at the time of germination and those in inner cortical tissues were more strongly transverse by 24HPG.

By 65HPG, when hypocotyl cells were rapidly elongating, the outer epidermal face exhibited a MT angle trend towards axial alignment (*Figure 2A,B*; ~32% between 80–90°). It is possible that this was due to the upcoming growth arrest these cells would soon experience. It is equally probable that this is the general angle trend seen on the outer epidermal face in the early stage of elongation at the hypocotyl base. The inner epidermal face of 65HPG cells did show MT signal and exhibited an transverse angle distribution (*Figure 2A,B*; ~44% between 0 and 10°). These data led us to conclude that during anisotropic growth from the time of germination, MT-based anisotropy information likely came from cortex cells or inner epidermal faces. This conclusion is consistent with analyses in older hypocotyls (*Crowell et al., 2011*; *Peaucelle et al., 2015*). While it is difficult to compare values across experiments and imaging conditions, our values for percent microtubules presenting 'transverse angles' were weak compared to those reported previously (*Crowell et al., 2011*). This may mean that at these early stages, anisotropy from MTs is weak and only consolidates later through mechanical feedback (*Hamant et al., 2008*; *Sampathkumar et al., 2014*); however, we note that there is no quantitative functional data relating the degree of anisotropic growth, the degree of MT alignment, and the dynamic variability of these parameters. An attractive hypothesis is that in early dark-grown hypocotyl elongation, MT-based anisotropy and pectin-based elastic asymmetry work cooperatively to regulate anisotropy.

## Pectin chemistry and wall elasticity are asymmetric in the epidermis of the dark-grown hypocotyl starting at germination

Elastic asymmetry in hypocotyl epidermal cells is proposed to regulate anisotropic growth and has been attributed to the presence of more calcium cross-linked homogalacturonan (HG) epitopes in elongating walls (*Peaucelle et al., 2015*). Since it was unclear how calcium cross-linked HG might facilitate either increased elasticity or

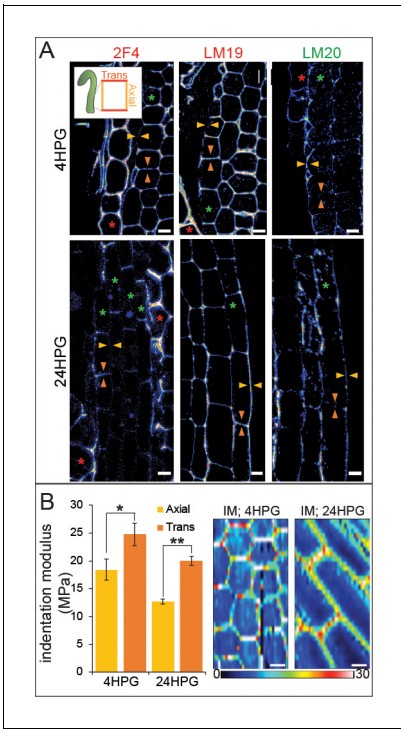

**Figure 3.** Changes in pectin chemistry underlie cell-level elastic asymmetry from the time of germination. (A) Representative longitudinal immunolocalization images of pectin methylation state as determined by LM19 (low degree of methylation) and LM20 (high degree of methylation) antibodies at 4HPG and 24HPG. Transverse and axial walls are indicated by orange and gold arrowheads, respectively (inset). Endosperm cells indicated by red asterisk, epidermal cell files by green asterisks. Negative controls for immunolocalizations can be found in *Figure 3—figure supplement 1*. (B) Representative maps of indentation moduli (IM; MPa) from base cells with representative graph of axial vs transverse IM for hypocotyls, at 4HPG and 24HPG (all replicate data can be found in *Figure 3—figure supplement 1*); Wilcoxan rank-sum test for significance: single asterisk, p<0.005; double asterisk, p<0.001. Scale bars = 10 μm.
DOI: https://doi.org/10.7554/eLife.38161.007

The following figure supplement is available for figure 3:

**Figure supplement 1.** Immunolocalization controls, IM replicates, cryo-SEM.
DOI: https://doi.org/10.7554/eLife.38161.008

increased growth, we undertook a broader examination of HG biochemistry on an expanded time frame. We performed cell-wall immunolocalizations on longitudinal sections of 4HPG and 24HPG hypocotyls to determine the distribution of methylated HG, de-methylated HG, and calcium cross-linked HG (using LM20, LM19, and 2F4 antibodies respectively). At 4HPG, slower-growing epidermal transverse walls were marked by 2F4 and LM19 indicating the presence of de-methylated HG and calcium cross-linked HG (*Figure 3A*, orange arrow heads; see inset for naming convention). The endosperm at this stage was also highly marked consistent with the literature (*Müller et al., 2013*) but without asymmetry (*Figure 3A*, red asterisk). In epidermal cells, the faster growing axial walls were marked with LM20 indicating the presence of methylated pectin (*Figure 3A*, gold arrow heads). At 24HPG, the asymmetry in 2F4 and LM20 antibodies was maintained, but the LM19 antibody marked both axial and transverse walls (*Figure 3A*; immunolocalization controls can be found in *Figure 3—figure supplement 1*). The lack of asymmetry in LM19 signal at 24HPG may indicate that as pectin is newly deposited it is de-esterified but only cross-linked in transverse walls; de-methylated HG in axial walls may be degraded (*Rui et al., 2017*; *Xiao et al., 2014*). When combined with the asymmetry in methylated HG signal (LM20), which could be due to differential delivery, these data hint at a complex cellular delivery mechanism. We concluded that slowly growing transverse walls had more de-methylated pectin which was calcium cross-linked, while faster growing axial walls had more methylated pectin.

We detected asymmetry in pectin biochemistry at 4HPG, consistent with our growth data indicating that hypocotyl cells were always growing anisotropically (*Figure 1*). A recent study postulated that a reported transition from isotropic to anisotropic growth in dark-grown *Arabidopsis* hypocotyl cells was due to the appearance of a cell-level elastic asymmetry around 15HPG (*Peaucelle et al., 2015*). To investigate cell wall elasticity under our conditions, where growth was anisotropic from germination onwards, we performed AFM-based nano-indentation on basal hypocotyl epidermides from dark-grown seedlings at 4HPG and 24HPG. At 4HPG, axial walls were more elastic (lower indentation modulus (IM)) when compared with transverse walls (18.4 MPa ± 1.9 vs 24.8 MPa ± 2.0; *Figure 3B*, *Figure 3—figure supplement 1*). This correlated well with our observations that cells at this early time point were growing anisotropically and presented asymmetric pectin epitopes (*Figure 1BC*, *Figure 1—figure supplement 1*). This difference was still observed in basal cells at 24HPG coincident with an increase in overall elasticity when they were entering into the rapid growth phase (12.7 MPa ±0.4 vs 20.0 MPa ±0.8; *Figure 3B*; *Figure 3—figure supplement 1*). At 24HPG, cell wall thickness was not significantly different between axial and transverse walls of basal hypocotyl cells, indicating that elasticity difference were underlain by biochemical and not geometrical differences (*Figure 3—figure supplement 1*). The elastic asymmetry increased to a ratio of 2 by 48HPG (*Figure 3—figure supplement 1*). It is possible that the increase in overall elasticity contributed to the shift to rapid growth observed at 24HPG and the start of the acropetal wave (*Figure 1*). From these data, it was apparent that a cell-level asymmetry in wall elasticity was present from the time of germination, coincident with growth anisotropy, and correlated with changes in pectin chemistry in dark-grown hypocotyl basal epidermal cells.

## In silico cell-level elastic asymmetry can increase growth anisotropy when combined with microtubule-based anisotropy

It has been proposed that pectin asymmetry alone might account for a shift to anisotropic growth. To test whether pectin asymmetry could induce anisotropic growth we next performed an in silico test. We developed a finite element method (FEM) model of a hypocotyl epidermis (based on methods in [*Bozorg et al., 2014*, *Bozorg et al., 2016*]). The FEM model consisted of a 3D epidermal layer made up of individual cells whose individual walls could have separate mechanical properties proscribed (*Figure 4A*; Appendix). Wall thickness was set according to the literature and our SEM observations (*Derbyshire et al., 2007a*). The epidermal layer was pressurized to provide the driving force for the growth of these cells and the space internal to the epidermis was also pressurized to simulate internal tissue force. Growth was implemented using a Lockhart model (*Lockhart, 1965*) where strain above a yield threshold set the growth rates relative to principal strain directions (*Bozorg et al., 2016*). Expanded details of the model can be found in the Appendix.

When the axial and transverse anticlinal walls had the same elasticity (no asymmetry) and when no material anisotropy was specified (cellulose orientation was not coordinated), the pressure forces caused the maximal strain (and stress) to be transverse (*Figure 4A*). This result would lead to a radial

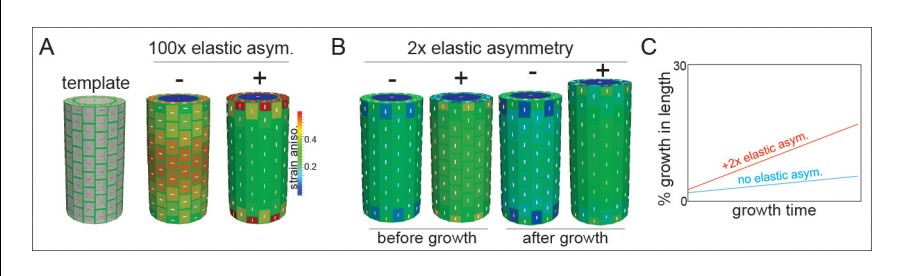

**Figure 4.** Microtubule alignment and cell elastic asymmetry additively regulate anisotropic growth in silico. (**A**) Template for finite element method simulation of a growing hypocotyl epidermis, alongside predicted strain anisotropy (growth) with no anisotropic information and with or without 100x elastic asymmetry added. White lines indicate the maximal stress direction. (**B**) Pre-growth and after-growth simulation results for a case with material anisotropy information provided by the internal epidermal face. The magnitude of anisotropy is enhanced by the addition of 2-fold elastic asymmetry, resulting in increased growth of the template. (**C**) Simulation output as percent growth in time from the simulation in (**B**) showing the increased relative growth achieved by addition of 2-fold elastic asymmetry. Simulation result from simulated internal tissue (cortex) anisotropy can be found in *Figure 4—figure supplement 1* alongside results of sensitivity analyses for both simulations.
DOI: https://doi.org/10.7554/eLife.38161.009

The following figure supplement is available for figure 4:

**Figure supplement 1.** Secondary simulation and sensetivity analyses.
DOI: https://doi.org/10.7554/eLife.38161.010

swelling of the organ and was consistent with basic mechanical theories of hoop stress (*Baskin and Jensen, 2013*). In order to drive axial anisotropy, in the absence of cellulose-based material anisotropy, a 100-fold elastic asymmetry had to be invoked (*Figure 4A*). These results led us to hypothesize that a 2-fold elastic asymmetry alone, as measured in our experiments (*Figure 3*), would be insufficient to drive anisotropic growth.

Based on the literature, and our own observations of MT angle at early growth stages, we next added material anisotropy to our simulations. When material anisotropy favoring axial strain was specified at the inner epidermal wall (as measured in [*Crowell et al., 2011*]), we obtained axial growth anisotropy (*Figure 4B*); strikingly, addition of a 2-fold elastic asymmetry, consistent with our experiments (*Figure 3*), enhanced the magnitude of growth anisotropy (*Figure 4B,C*). Since it was also possible that internal tissue provided anisotropic information (i.e. the cortex, [*Hejnowicz et al., 2000*]), we simulated this situation by specifying axial pressure in the inner-epidermal space and also recovered axial anisotropy (*Figure 4—figure supplement 1*). The addition of 2-fold elastic asymmetry in the epidermis was again able to enhance the magnitude of axial anisotropy (*Figure 4—figure supplement 1*). A sensitivity analysis of the two cases with anisotropic information and 2-fold elastic asymmetry indicated that they were most sensitive to variation in the degree of anisotropy and that increasing elastic asymmetry showed a positive correlation with growth anisotropy (*Figure 4—figure supplement 1*). In conclusion, our finite element mechanical model led us to propose that while epidermal elastic asymmetry alone was insufficient to drive axial growth anisotropy, it was able to contribute by increasing the anisotropy achieved when anisotropic information was provided by inner tissues or the inner epidermal wall.

## Ectopic changes in pectin biochemistry alter cell anisotropy and organ growth

Our experimental and computational results led us to believe that pectin biochemistry could have an impact on growth anisotropy; however, our observations were correlative. In order to test a causal relationship, we altered pectin methylation in dark-growing hypocotyls and observed any subsequent changes in the cell shape. In *Arabidopsis*, the methylation of HG can be controlled by the antagonistic activity of two protein families, PECTIN METHYLESTERASE (PME) and PECTIN METHYLESTERASE INHIBITOR (PMEI); PME activity leads to de-esterification and likely to calcium cross-linking and increased rigidity, while PMEI would have the opposite effect (*Caffall and Mohnen,*

*2009*; *Levesque-Tremblay et al., 2015a*, *Levesque-Tremblay et al., 2015b*). Note that PME activity could also lead to HG degradation by POLYGALACTURONASE (PG), whose activity is also important for proper hypocotyl growth (*Rui et al., 2017*; *Xiao et al., 2014*).

To alter pectin methylation in the hypocotyl, we utilized transgenic lines expressing either PECTIN METHYLESTERASE5 (PME5) or PECTIN METHYLESTERASE INHIBITOR3 (PMEI3) under ethanol induction (*Peaucelle et al., 2008*) (Verification of induction in *Figure 5—figure supplement 1*). We used AFM-based nano-indentation to examine basal cell wall elasticity in induced hypocotyls (non-transgenic (NT), PME, and PMEI). We observed that both PME5 and PMEI3 induction abolished cell-level elastic asymmetry: PME5 increased the rigidity in both axial and transverse anticlinal walls, while PMEI3 decreased the rigidity in both (*Figure 5A*; *Figure 5—figure supplement 1*). These changes in cell wall elastic asymmetry were accompanied by changes in cell shape anisotropy: induction of PMEI3 led to more anisotropic cells within the elongation wave and PME5 induction to less anisotropic cells (*Figure 5B*; *Figure 5—figure supplement 1*). Note that the position of the wave was not altered with PMEI3 induction, indicating that ectopically altering pectin chemistry could alter growth rate but not the position of the acropetal wave. Also, anisotropy was not lost in either transgenic induction indicating that loss of cell wall asymmetry alone is not enough to abolish anisotropic cell shape, or presumably growth. Altogether, it appears that pectin asymmetry has a contributory, not sole regulatory, role in anisotropy supporting our computational results.

When NT, PME5 and PMEI3 hypocotyls were exposed to the inducer the following changes in growth were observed at the organ level: when compared to NT, PME5 induction abolished the rapid elongation phase and PMEI3 induction increased early elongation essentially flattening out the difference between the slow and rapid phases (*Figure 5C*). These results were consistent with a promotive role for pectin methylation in rapid hypocotyl elongation. To confirm that the expected changes in pectin chemistry were occurring, we performed immunolocalizations on transverse

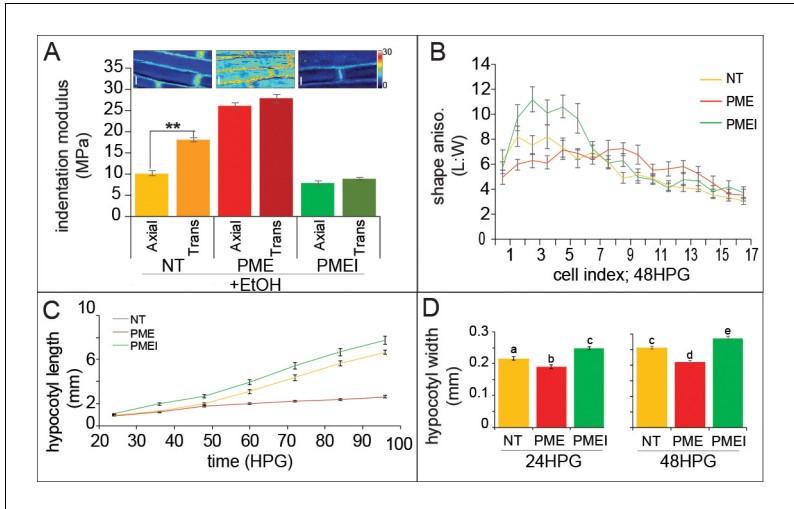

**Figure 5.** Ectopic alteration of pectin biochemistry alters cell anisotropy and hypocotyl elongation. (A) Indentation modulus (IM) for ethanol treated non-transgenic (NT, gold), *alcA::PME5* (PME, red) and *alcA::PMEI3* (PMEI, green) basal hypocotyl cells at 48HPG split into axial and transverse walls; Wilcoxan rank-sum test for significance: double asterisk, p<0.001. Full data set can be found in *Figure 5—figure supplement 1*. Scale bars = 10 µm. (B) Shape anisotropy (length:width) of cells by position index in NT, PME and PMEI induced seedlings after 48H. See *Figure 1* for position indexing. Induction controls and cell level length and width measurements can be found in *Figure 5—figure supplement 1*. (C) Hypocotyl length at discrete time points, extracted from infrared imaging of hypocotyl growth over time, for induced NT, PME, and PMEI seedlings. At p<0.05 (t-test) all data points in (C) are significantly different except PME and NT at 36HPG and 48HPG. (D) Hypocotyl width of induced NT, PME and PMEI seedlings at 24HPG and 48HPG. a,b,c indicate statistical similarity based on pairwise t-tests (p<0.001).
DOI: https://doi.org/10.7554/eLife.38161.011

The following figure supplement is available for figure 5:

**Figure supplement 1.** IM, immunolocalizations and induction verification.
DOI: https://doi.org/10.7554/eLife.38161.012

sections of hypocotyls; namely, that PME5 induction yielded more de-methylated pectin signal (LM19 antibody; *Figure 5—figure supplement 1*) and that PMEI3 induction yielded more methylated pectin signal (LM20 antibody; *Figure 5—figure supplement 1*). These data are thoroughly consistent with an increase in the relative amount of pectin methylation contributing to the transition from slow to rapid elongation, a point we will revisit once again at the end of this report.

As we were primarily interested in anisotropic growth, we also examined how hypocotyl width was altered with changes in pectin biochemistry. Commensurate with the change in cell-level anisotropy, PME5 induction resulted in a reduced hypocotyl length (*Figure 5C*) and also a reduction in hypocotyl width (*Figures 5D*, 24 and 48HPG). Conversely, induction of PMEI3 led to an increase in hypocotyl length (*Figure 5C*) and an increase in hypocotyl width (*Figures 5D*, 24 and 48HPG). Taken together these data hint at a role for pectin chemistry in cell, and organ, growth anisotropy; however, in no case was anisotropy completely abolished indicating a more complex regulation of anisotropy than pectin asymmetry alone could provide, further supporting our additive model for anisotropy in the hypocotyl.

## Ectopic alterations in pectin biochemistry can mediate the effect of microtubule disruption

Our observations of weak MT transverse alignment and pectin asymmetry, and our computational modelling, strongly indicated an additive role for these two mechanical factors. Since we had observed that ectopic alteration of pectin biochemistry could not fully abolish cell-level anisotropy, we next asked whether loss of MT-based anisotropy could be affected by altering pectin. Oryzalin, a drug that blocks the polymerization of MT, is known to affect the trajectories, distribution and densities of cellulose synthase complexes (*Paredez et al., 2006*; *Chan et al., 2010*), to change the organization in cellulose microfibril orientations, and to induce cell swelling (a trend towards isotropy) (*Baskin et al., 2004*, *Baskin et al., 1994*; *Lucas et al., 2011*). We treated seedlings with 5 μM oryzalin while inducing either PME5 or PMEI3. NT control hypocotyls, treated with the inducer, showed a reduction in cell shape anisotropy when treated with oryzalin indicating typical cell swelling (*Figure 6A,B*). This effect was reduced in induced PME5 plants and enhanced in induced PMEI3 plants (*Figure 6A,B*). While there was a response to oryzalin in PME5-induced hypocotyls, the cell swelling was reduced indicating a compensatory cell wall strength in these cells (*Figure 6B*). The opposite was true in oryzalin-treated PMEI3-induced hypocotyls where the cell swelling was more dramatic than either PME5 or NT, despite induced-PMEI3 cells being

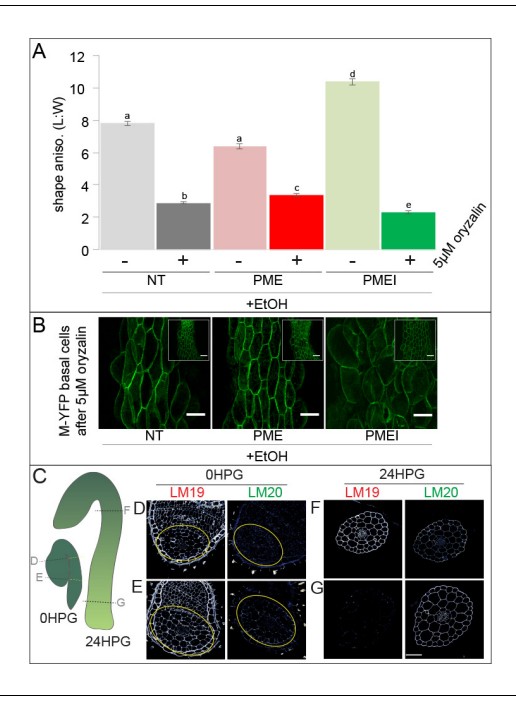

**Figure 6.** Pectin biochemistry contributes to wall mechanical strength and its changes correlate with the acropetal wave. (**A**) Shape anisotropy (length/width) of cells at the bottom of treated and control hypocotyls. Error bars represent the standard error of the mean. All seedlings were treated with the inducer (ethanol; EtOH). Scale bars = 100 μm. a,b,c,d,e indicate similarity based on pairwise t-tests (p<0.005). (**B**) Representative images of basal cells from NT, PME5 and PMEI3 induced hypocotyls expressing the Myr-YFP membrane marker grown for 48 hr on media containing 5 μM oryzalin. (**C**) Diagrammatic representation of 0HPG and 24HPG hypocotyls showing the relative positions of transverse sections in D-G. (**D–G**) Representative immunolocalizations on transverse sections of 0HPG and 24HPG hypocotyls for de-methylated and methylated pectin (LM19 and LM20 respectively). Yellow lines (**C**) and circles (**D,E**) demark the hypocotyl where sections included cotyledons. Scale bar = 50 μm. 2F4 immunolocalization at 24HPG may be found in *Figure 6—figure supplement 1*.

DOI: https://doi.org/10.7554/eLife.38161.013

The following figure supplement is available for figure 6:

**Figure supplement 1.** 2F4 immunolocalization at 24HPG.

DOI: https://doi.org/10.7554/eLife.38161.014

the most anisotropic without oryzalin treatment (*Figure 6A,B*). These data indicated that changes in pectin biochemistry could modulate the effect of MT-derived cell swelling and isotropy, but again this modulation was never complete. Pharmacological treatments would be unlikely to affect existing cellulose fiber alignment from before the time of treatment, and so it is likely that treated hypocotyl cell walls maintained some pre-treatment cellulose-based anisotropy.

We observed that the slowly-growing upper regions of dark-grown hypocotyls in these experiments exhibited less swelling upon oryzalin treatment (*Figure 6B*, insets). Taken together with our earlier observations of decreasing pectin-based wall IM over time (*Figure 3B*; *Figure 3—figure supplement 1*) and the effect of ectopic pectin alteration on hypocotyl elongation (*Figure 5C*), we wondered if there might be endogenous differences in pectin chemistry along the hypocotyl length. To examine pectin biochemistry in this context, we performed immunolocalizations on transverse sections of dark-grown hypocotyl at 0HPG, when all parts of the hypocotyl were slowly growing, and 24HPG, when basal cells were entering the rapid elongation phase, using antibodies for methylated HG (LM20) and de-methylated HG (LM19). At the time of germination (0HPG) we observed strong signal for de-methylated HG (LM19) in basal and apical transverse sections (*Figure 6D,E*) and weak signal for methylated HG (LM20; *Figure 6D,E*). At 24HPG, the de-methylated HG signal remained high in the slow-growing apical sections but was low in basal sections (*Figure 6F,G*). The calcium cross-linked HG antibody 2F4 showed a similar pattern (*Figure 6—figure supplement 1*). Methylated HG exhibited a complementary pattern with lower signal in sections from slow growing apical regions and higher signal in sections from rapidly growing basal regions (*Figure 6F,G*). We could not discern any difference between tissue layers in our sections, indicating an organ-wide change in HG methylation state. These data led us to hypothesize that de-methylated pectin kept the hypocotyl in a slowly growing state, while methylated pectin allowed it to grow rapidly. An attractive hypothesis is that maintenance of pectin methylation allows for the onset of the acropetal growth wave; however, other growth-related parameters might be involved, such as vacuolar structure and resulting water uptake ability (*Scheuring et al., 2016*).

## Discussion

### The origin of material anisotropy

In multicellular anisotropically growing organs there is no reason *stricto senso* for every cell to have anisotropic wall properties (*Baskin and Jensen, 2013*). Indeed, our data presented here and those of others suggest that this is not the case in the dark-growing *Arabidopsis* hypocotyl (*Peaucelle et al., 2015*; *Pelletier et al., 2010*; *Derbyshire et al., 2007a*). Instead, it appears that anisotropic information originates at the inner face of epidermal cells and/or within cortical cells. Here we present a harmonious model of cell-wall controlled anisotropic growth: pectin asymmetry in the epidermis enhances anisotropic growth controlled by cellulose anisotropy.

Our experimental analysis of native pectin biochemistry and manipulations of pectin biochemistry support a role for pectin asymmetry within the epidermis as a contributor to anisotropic growth; axial epidermal walls present markers of a more elastic pectin matrix and these walls grow faster, while the slow-growing transverse cell walls present markers of more rigid pectin. These biochemical observations are backed up by parsimonious observations of cell wall elasticity. When native pectin biochemistry was over-ridden by ectopic expression of PME5 or PMEI3, native pectin asymmetry was an important component of anisotropic growth, but not the sole regulator. Our modelling experiments support an additive role for pectin asymmetry to anisotropic hypocotyl elongation, when combined with cellulose-based material anisotropy.

There is an elephant in the room: although a strong correlation between MT orientation, CESA track movements and/or cellulose microfibril orientation have been reported in the literature in several systems (*Crowell et al., 2011*; *Mueller and Brown, 1982*; *Takeda and Shibaoka, 1981*) there are also reports that show no correlation (*Sugimoto, 2003*; *Emons et al., 1992*, *Emons et al., 2007*; *Fujita et al., 2011*). In the latter, cellulose synthase complex (CSC) movements have been shown to persist even in the absence of MTs. In the *mor1-1* mutant, temperature-induced microtubule disorganization had no effect on cellulose microfibril orientation on inner epidermal walls (*Fujita et al., 2011*). It is also prudent to note that cellulose microfibrils may undergo passive alignment once deposited within the apoplast and as such MT and CESA orientations may not accurately reflect

cellulose fiber orientation (*Braybrook, 2017*). Given the observations and hypotheses above, it is possible that the outer epidermal wall of young dark-grown hypocotyls does contain transversely aligned cellulose fibrils in spite of the disperse orientation in both MT and CESA markers; however, direct imaging of cellulose fibers in hypocotyls just after germination is technically impossible at this time. In spite of this limitation, we believe the conclusion that pectin asymmetry contributes to aniso-tropic growth remains strong.

## Epidermal cell growth in the elongating, etiolated, hypocotyl

We demonstrate that hypocotyl epidermal cells are anisotropic from the time of germination, an observation made possible by measuring both cell length and cell width. These observations build upon work describing the changes in cell length alone during elongation and the definition of the acropetal wave (*Gendreau et al., 1997*). By assessing cell width and length in time we have been able to proxy each cell's anisotropic growth. We further hypothesize that this early anisotropic growth is likely directed by internal wall material anisotropy (inner epidermal face, cortex walls or their combined weak material anisotropy) and is enhanced by cell-level elastic asymmetry.

Our *in silico* modelling approaches have allowed us to further explore our hypotheses and pro-vided some insight into their validity. First, the measured elastic asymmetry (2-fold, consistent with that reported recently (*Peaucelle et al., 2015*) was insufficient to drive axial anisotropic growth in our epidermal model, unless accompanied by internally provided anisotropic force or anisotropic properties of the inner epidermal wall. When combined, a 2-fold elastic asymmetry enhanced the anisotropic growth directed by internal tissues, leading us to hypothesize that elastic asymmetry aids in growth anisotropy. It is possible that the difference measured by our indentation tests underesti-mated the effective elasticity of the hypocotyl cell walls. It may also be that there are internal mechanical pectin asymmetries which contribute to anisotropy; our current indentation methods are restricted to epidermal cells. Cell wall elasticity, as measured here, is also an immediate property of cell walls which is only correlated with growth; future work must focus on uncovering how cell wall elasticity might relate to cell wall growth (*Braybrook and Jönsson, 2016*; *Braybrook, 2015*).

## Towards a functional understanding of pectin biochemistry and the cell wall

Our data supports a role for pectin methylation in rapid cell elongation in the dark-grown *Arabidop-sis* hypocotyl: pectin methylation is high in rapidly elongating hypocotyl cells as is wall elasticity; when pectin methylation is enhanced, walls are more elastic and rapid elongation starts early; when pectin de-methylation is induced, walls are less elastic and the rapid elongation phase is suppressed.

An attractive hypothesis would be that maintenance of newly deposited pectin in a methylated state allows cell walls to expand more rapidly, and the conversion to a de-methylated state slows growth in the dark-grown *Arabidopsis* hypocotyl. Pectin with low methylation has been associated with non-growing areas in different species (*Fenwick et al., 1997*; *Fujino and Itoh, 1998*; *Liber-man, 1999*). However, the literature is complex: PMEI activity (maintenance of methylated pectin) has been associated with increased cell elongation in the *Arabidopsis* root (PMEI1 and PMEI2 over-expression [*Lionetti et al., 2007*]), but expression of another PMEI (PMEI4) in the hypocotyl delayed the transition to rapid growth and had no effect on length growth rate (*Pelletier et al., 2010*); PMEI3 induction generated stiffer walls in *Arabidopsis* shoot meristems (*Braybrook and Peaucelle, 2013*; *Peaucelle et al., 2011*). There are several possible explanations for these differences in phe-notype: since the PMEI family is large and diverse (*Wang et al., 2013*), it is likely that different pro-teins have different activities due to structure and environment (*Bou Daher and Braybrook, 2015*; *Giovannoni, 1989*; *Jolie et al., 2010*; *Tian et al., 2006*); as we have demonstrated here, analysis of cell or organ length alone may obscure changes in width and it is possible that PMEI4 induced greater but more isotropic growth. When *PMEI5* was overexpressed in adult *Arabidopsis* plants, stems displayed twice the diameter compared to controls further supporting the need to examine both width and length of organs (*Müller et al., 2013*).

The PME over-expression literature is less complicated (possibly due to being slimmer): ectopic PME expression has been shown to reduce methylation and hypocotyl length previously, consistent with our study (*Derbyshire et al., 2007b*). However, a hypocotyl-expressed PME (At3G49220) was found to be highly expressed after 30HPG and in elongating cell regions (*Pelletier et al., 2010*).

Our data show that both methylated and de-methylated HG can be observed in rapidly elongating hypocotyls (*Figures 3* and *6*) indicating that PME activity is converting newly deposited methylated pectin into a de-methylated state during elongation. However, we must note that the available antibodies do not discriminate between patterns or degree of de-methylation. Different PMEs may have different activities (*Wolf and Greiner, 2012*) resulting in HG chains more likely to cross-link or be targeted for polygalacturonase-mediated degradation, fates which may be linked to the pattern of de-methylation (*Wakabayashi et al., 2000*, *Wakabayashi et al., 2003*). As an example, knockdown or silencing of a pollen PME in tobacco and *Arabidopsis* led to reduced pollen tube elongation (*Bosch and Hepler, 2006*; *Jiang, 2005*) but treatment of the pollen tubes with orange-peel PME also reduced growth (*Marc et al., 1998*); it is not clear whether these apparently contradictory results are due to differential PME activity but the situation is clearly complex. PMEs also exhibit differential activities due to pH (*Jolie et al., 2010*; *Hocq et al., 2017*). While PMEs with alkaline pI remove the methyl groups in blocks, acid pI PMEs do so in a random fashion (*Jolie et al., 2010*). Most of the *Arabidopsis* PMEs have an alkaline pI but there are some with acidic pI (*Tian et al., 2006*). The pattern of de-methylation has an impact on the fate of HG with block-wise de-methylation leading to Ca-cross linking and random leading to degradation by PG (*Willats et al., 2001*). While some data exists for transcriptional changes in PME and PMEI genes (*Pelletier et al., 2010*) it remains to be seen whether these changes result in changes in wall biochemistry and mechanics given the complexities of their post-translational activities.

Perhaps the most puzzling contradiction to our data is the opposite phenotype shown recently for cell shape and rigidity for the same transgenic lines (*Peaucelle et al., 2015*); in our work we see a full (100%) penetrance of phenotype upon induction (*Figure 2—figure supplement 1*), whereas the earlier study reported only a 10–20% penetrance of their desired phenotype – seedlings which were carried through for further analyses (*Peaucelle et al., 2015*). It is possible that the amount of induction stimulus, or delivery method, had an effect on the phenotype presented; however, we must note that our immunolocalizations and rigidity data are consistent with the predicted biochemical activity of both PME and PMEIs. We used two markers for opposing states of pectin biochemistry, whereas only looking at de-methylated pectin alone might have given a reduced picture. Our analyses of cell shape upon microtubule disruption (oryzalin treatment) combined with pectin biochemistry manipulation further support our mechanical interpretation of the pectin-mechanics relationship put forward here: ectopic PME5 expression leads to stronger cell walls and partially suppresses oryzalin-induced cell swelling while ectopic PMEI3 expression enhances cell swelling and isotropy. Taken together, our data suggest an important role for pectin methylation in hypocotyl anisotropy; they also highlight the complexity of the experiments and the field as a whole.

Lastly, feedback between cell-wall integrity and wall biochemistry/structure may add more complexity to the system as the oligogalacturonides generated by the lysis of the HG can act as signaling molecules and affect plant development (for more references and reviews see [*Wolf and Greiner, 2012*; *Wolf et al., 2009*]). In fact, these oligogalacturonides have been recently shown to be responsible for sustaining cell elongation in dark grown hypocotyls (*Sinclair et al., 2017*). Due to the overexpression system used here, an ethanol induced transcriptional system, there is a time lag between induction and response. During this time and growth-time itself, we cannot discount that changes in wall biochemistry and mechanics induced by our PME and PMEI might have fed back through this system resulting in altered growth or further alterations in mechanics. Again though, we must stress the parsimonious nature of the predicted role of de-methylation on pectin gel mechanics, the observed mechanical changes in our system, the co-incident changes in wall biochemistry, and changes in cell shape and growth.

## Conclusions

We will return, lastly, to the question we began with: How does a seedling elongate upwards rapidly? The data presented here make a strong case that changes in pectin chemistry, and resultant wall rigidity, are important for the anisotropic growth that is critical for the hypocotyl. However, changes in pectin alone are likely insufficient to direct anisotropy: we observed that anisotropy of internal tissues is likely to be required for anisotropic growth which is aided by elastic asymmetry in the epidermis. We therefore present a harmonious model of dark-grown hypocotyl elongation where the anisotropy provided by cellulose is enhanced by epidermal elastic asymmetry. Our experiments and conclusions also leave us with several new questions: how does altered elasticity actually affect

growth? How might elastic asymmetry be established in the first place? How is the acropetal wave, and change in pectin chemistry, instructed? These are questions which we look forward to investigating in the future.

# Materials and methods

## Key resources table

| Reagent type (species) or resource | Designation | Source or reference | Identifiers | Additional information |
|---|---|---|---|---|
| Biological sample (*Arabidopsis thaliana*) | PMEI3 | PMID19097903 | | |
| Biological sample (*Arabidopsis thaliana*) | PME5 | PMID19097903 | | |
| Biological sample (*Arabidopsis thaliana*) | GL2::GFP | Nottingham Arabidopsis Stock Center (NASC) | ID_NASC: N66491 | |
| Biological sample (*Arabidopsis thaliana*) | 35S::GFP-MAP4 | PMID 9811799 | | |
| Biological sample (*Arabidopsis thaliana*) | 35S::GFP-TUA6 | doi:10.1007/BF01279267 | | |
| Biological sample (*Arabidopsis thaliana*) | 35S::GFP-EB1 | PMID 14557818;14614826 | | |
| Biological sample (*Arabidopsis thaliana*) | CESA3::CESA3-GFP | PMID 17878303 | | |
| Biological sample (*Arabidopsis thaliana*) | Ubq::MYR-YFP | PMID 27212401 | | |
| Antibody | LM19 | Plant Probes, UK | ID_PlantProbes: LM19; RRID: AB_2734788 | 1/200 dilution |
| Antibody | LM20 | Plant Probes, UK | ID_PlantProbes: LM20; RRID: AB_2734789 | 1/200 dilution |
| Antibody | 2 F-4 | other | | 2 F-4: P. van Custems (gift); 1/100 dilution |
| Antibody | DyLight 488 goat anti-rat | Cambridge Bioscience/ Bethyl, UK | ID_CamBioSci:A110-100D2; RRID: AB_10630108 | 1/400 dilution |
| Antibody | Goat Anti-Mouse IgG (H + L) Antibody Alexa Fluor 488 | Invitrogen, UK | ID_Invitrogen:A11017; RRID: AB_143160 | 1/200 dilution |
| Chemical compound, drug | oryzalin | SIGMA | ID_SIGMA: 36182 | |
| Software | MatLab2016a | MathWorks, Inc., USA | RRID:SCR_001622 | |
| Software | R3.4.1 | other | RRID:SCR_001905 | R3.4.1: https://www.r-project.org/ |
| Software | JPK SPM Data Processing software, v. spm 5.0.69 | JPK Instruments, DE | | |
| Software | MicroFilament Analyzer | PMID 23656865 | RRID:SCR_016411 | |
| Software | ImageJ/Fiji | PMID 22743772 | RRID:SCR_003070 | |

## Growth conditions

Transgenic lines sourced as indicated in Key Resources Table. Seeds were germinated on ½ MS plates containing Gamborg's B5-vitamins but no sucrose. Germination was defined as the time when the radicle broke through the endosperm (0HPG). At this time, seedlings were selected and aligned with the radicle pointing downwards on ½ MS+B vitamins with 1.5% sucrose. The plates were wrapped with two layers of foil to simulate constant darkness. For PME5 and PMEI3 (*Sampathkumar et al., 2014*) induction, 0HPG seedlings placed in the middle of petri dishes flanked by two 500 µl microfuge tubes were placed at each side containing 200 µl of 100% ethanol each; this treatment achieved 100% penetrance of phenotype (*Figure 5—figure supplement 1*). Orzyalin treatment was as follows: Oryzalin was dissolved in DMSO and added to cooled media before pouring, to a final concentration of 5 µM. Mock treatment consisted of DMSO; 0HPG seeds (genotype

PMEI3/M-YFP, PME5/M-YFP or M-YFP alone; F3 homozygous lines generated by crossing) were transferred to media with oryzalin or mock (DMSO) and grown in the dark for 48H prior to confocal imaging.

## Immunolabelling

Immunolocalizations were performed on 0.5 µm thick sections of LR White embedded hypocotyls. LM19 and LM20 antibodies (PlantProbes, UK) were diluted 200 times in PBS with 2% BSA. DyLight 488 goat anti-rat (Cambridge Bioscience/Bethyl) secondary antibody was diluted 400 times. 2F4 (P. van Cutsem, gift) immunolabelling was performed as in (*Liners et al., 1992*) Briefly, the primary antibody was diluted 100 times in TCN (20 mM Tris, 0.5 mM $CaCl_2$ and 150 mM NaCl) with 1% w/v skim milk. Alexa Fluor 488 goat anti-mouse (Invitrogen, UK) secondary antibody, was diluted 200 times. Images were acquired using a Leica TCS SP8 confocal microscope. Ratios were obtained using ImageJ by drawing a line along the walls in question and using the average fluorescence intensity of the line. Sample numbers were: Transverse 4HPG, n = 7 (from 2 hypocotyls); Transverse 24HPG, n = 12 (from 3 hypocotyls); ratio calculations LM19, n = 72 from 7 sections; ratio calculations LM20, n = 41 from 7 sections; transverse 0HPG, n = 6 (from 2 hypocotyls); transverse 24HPG, n = 14 (from 4 hypocotyls); control immunos, n = 9 (from 2 to 4 seedlings each); transverse 48HPG for NT/PME/ PMEI, n = 9 each (from 4 hypocotyls).

## GUS staining

Seedlings were incubated for 6 hr at 37°C in a solution of 50% water and 50% 2x GUS stain (50 mM $KPO_4$, 0.1% triton X-100, 0.3 mg/mL X-GlcA (5-Bromo-4-chloro-3-indolyl-β-D-glucuronic acid, sodium salt), 1 mM $K_4Fe(CN)_6$, 1 mM $K_3Fe(CN)_6$, 0.1 v/v 1M $KPO_4$ pH 7 (61.5mL 1M $K_2PO_4$ and 38.5 mL 1M $KH_2PO_4$ in 100 mL). samples were washed three times in 70% ethanol and one time in water and mounted in 50% glycerol under a coverslip and sealed with nailpolish.

## Microtubule and cellulose synthase complexes imaging and assessment

Images were acquired from *35S::GFP-MAP4*, *35S::GFP-TUA6*, *35S::GFP-EB1* and *CESA3::CESA3-GFP* seedlings using a Leica TCS SP8 confocal microscope using a 63X oil objective (1.4 numerical aperture). For microtubule orientation, we used the MicroFilament Analyzer (MFA) tool (*Jacques et al., 2013*). Sample numbers were: 0H: n = 65 (from 4 to 5 hypocotyls); 24H: n = 30 (from 9 hypocotyls); 65HPG: n = 13 (from 6 hypocotyls). For cortex analysis, n = 36 (from 5 hypocotyls).

## Cell growth and shape analyses

20 seedlings of *Arabidopsis thaliana* expressing a myristoylated-YFP were imaged for each time point, as confocal imaging stopped dark-grown hypocotyl elongation. Images were acquired using a Leica TCS SP8 confocal microscope. For cells, length and width were measured in Fiji (*Liners et al., 1992*); data were collected from 2 to 3 non-dividing files per hypocotyl. Cell diameter was recorded at the level of the central length of each cell. For cell shape analyses in induced NT, PME, and PMEI plants 10 seedlings of each were analysed for 48HPG and 24HPG. Hypocotyl widths and lengths at 24HPG and 48HPG were measured in 6–12 hypocotyls per treatment. Dividing cell characterization was conducted on 20 hypocotyls for each time point, and 4 seedlings were screened for *GL2::GFP* expression pattern. For oryzalin treated seedlings, a total of 12 seedlings for each treatment were imaged and the dimensions 8 cells per seedling, from the base, were measured.

## Infrared growth imaging and analysis

For imaging dark grown hypocotyls, a custom IR imaging setup was used, design available upon request. Images were acquired at 10 min intervals over 5 days. Images for selected time points were extracted and hypocotyl length was measured in Fiji (*Schindelin et al., 2012*).

## Atomic force microscopy

Further discussion of AFM methods and interpretation can be found in the Technical Supplement. AFM-based nano-indentation experiments were designed and performed according to (*Braybrook, 2015*). Briefly, dissected and plasmolyzed (0.55M mannitol; minimum 15 min) hypocotyls

were indented using a Nano Wizard 3 AFM (JPK Instruments, DE) mounted with a 5 nm diameter pyramidal indenter (Windsor Scientific, UK) on a cantilever of 45.5 N/m stiffness; cantilever stiffness was calibrated by thermal tuning. For each hypocotyl, two areas of 50 × 100 μm were indented with 16 × 32 points: an area just before the collet and one slightly higher, to encompass basal cells. Indentations were performed with 500nN of force yielding an indentation depth range of 250–500 nm. Sample numbers were as follows: 4HPG, n = 24 cells (from 6 hypocotyls); 24HPG, n = 18 cells (from 6 hypocotyls); PME/PMEI/NT at 48HPG, n = 18 cells (from 6 hypocotyls each). Force indentation approach curves were analyzed using JPK SPM Data Processing software (JPK Instruments, DE; v. spm 5.0.69) using a Hertzian indentation model and a pyramidal tip shape. We have chosen to adopt the term 'indentation modulus' in place of 'Young's Modulus' or 'Apparent Young's Modulus' in order to distinguish these tests from those designed to assess Young's modulus in materials science (*Cosgrove, 2016*). Indentation modulus maps were then imported into MatLab (MATLAB 2016a, MathWorks, Inc., USA) and values were selected from anticlinal cell walls. For each grid area, 10–50 points were chosen from anticlinal walls and used for subsequent analyses, representing data from 3 to 10 cells depending on cell length in the scan area. Rations of IM were calculated by straightforward division of averages and propagation of SEM. Mann-Whitney tests for significant differences were performed in R as distributions were non-normal. A technical discussion on AFM-based analyses may be found in the Appendix.

## CryoSEM

Brass stubs were covered with 50% lanolin solution in water that was preheated to 50°C and vigorously vortexed prior to applying the seedlings. 24HPG seedlings were placed on the lanolin coat and immediately plunge frozen in liquid nitrogen under vacuum. Frozen samples were then transferred under vacuum to a prep chamber of a PT3010T cryo-apparatus (Quorum Technologies, Lewes, UK) and maintained at −145°C. For cryo fracture a level semi-rotary cryo knife was used to randomly fracture the hypocotyls. All samples were sputter coated with a 3 nm platinum coat. Samples were then transferred and maintained cold under vacuum into the chamber of a Zeiss EVO HD15 SEM fitted with a cryo-stage. Images were taken on the SEM using a gun voltage of 6 kV, I probe size of 460 pA, a SE detector and a working distance of 4 mm.

## Computational modelling

Details of the modelling can be found in the Appendix. In brief, a 3D finite element methods mechanical model was developed to evaluate mechanical signals and growth for cell walls of the epidermal cell layer of a hypocotyl. We used a 3D template and where prisms with six walls were utilized to represent individual cells (*Figure 4*). Each wall was triangulated from its centroid into triangular (planar) elements. The dimensions of the cell walls are proportional to the average values of those seen in experiments, for example *Figure 1*. The thickness of the walls was included by adjusting their corresponding Young's moduli assuming the material strength is proportional to the amount of material in a unit area. Individual cells were assembled into a 3D structure representing an epidermal cell layer. A wall in between two cells was divided into two adjacent walls and connected via the corner nodes. In this set up each pair of adjacent walls experienced the same deformation while they could hold individual mechanical properties. The two ends of the template were closed and the tissue was pressurized on the outer surface. The mechanical signals of cells close to each end were excluded from the analysis to avoid artefacts caused by boundary conditions (see simulation edges in *Figure 4*). In order to reduce the boundary effects, vertices at the two ends of the cylinder were constrained to stay in a plane parallel to the XY plane while allowed to move freely in the X and Y directions.

## Statistical information

For all of our analyses we did not exclude any data points. For AFM-based experiments samples sizes were low due to technical difficulty in experimentation: sample mounting was very difficult and often of 10 mounted samples only 2 remained fixed at 4HPG and 24HPG. AFM-based data was non-normally distributed so a Wilcoxan rank-sum test (aka Mann-Whitney-Wilcoxan) was used to see if the data from two independent samples were equivalent: this nonparametric test follows the null hypothesis that a random value selected from group 1 is equally likely to be greater or lesser than a

random member of group 2. For normally distributed data, such as growth and cell dimensions, t-tests were used (singly or pair-wise comparisons); for multi-sample comparisons (e.g. NT vs. PME vs. PMEI) pair-wise t-tests are shown but ANOVA gave similar results.

## Data and code availability

All raw data produced and utilized in this study can be downloaded from the DRYAD data repository (doi:10.5061/dryad.4s4b3nf). Modeling code can be accessed through the Sainsbury Laboratory's GitLab page (https://gitlab.com/slcu/teamHJ/behruz/3Dhypocotyl; copy archived at https://github.com/elifesciences-publications/3Dhypocotyl).

## Acknowledgments

The authors thank Dr. R Wightman, Dr. S Costa, and Dr. O Hamant for gifts of seeds and Dr. P van Cutsems for gift of the 2F4 antibody. We thank Dr. S Schornack for figure design consultation, Prof. A Fleming and Dr. S Robinson for critical comments on the manuscript. Dr. A Peaucelle is thanked for his establishment of the plant-AFM method. Work in the Braybrook group in Cambridge was funded by The Gatsby Charitable Foundation (GAT3396/PR4), the BBSRC (BB.L002884.1), and a Marie Curie Actions CIG (No. 631914). Work in the Braybrook group at UCLA is funded by The Department of Cell, Molecular and Developmental Biology and the College of Life Sciences. Work in the Jönsson group is funded by The Gatsby Charitable Foundation (GAT3395/PR4), the Swedish Research Council (VR2013-4632), the Knut and Alice Wallenberg Foundation via ShapeSystems.

## Additional information

### Funding

| Funder | Grant reference number | Author |
| --- | --- | --- |
| Biotechnology and Biological Sciences Research Council | BB.L002884.1 | Firas Bou Daher Yuanjie Chen Jack Clough |
| Horizon 2020 Framework Programme | 631914 | Siobhan A Braybrook |
| Knut och Alice Wallenbergs Stiftelse | ShapeSystems | Behruz Bozorg Henrik Jönsson |
| University of California, Los Angeles | 403976-SB-69313 | Firas Bou Daher |
| Gatsby Charitable Foundation | GAT3396/PR4 | Firas Bou Daher Yuanjie Chen Jack Clough |
| Svenska Forskningsrådet Formas | VR2013-4632 | Behruz Bozorg Henrik Jönsson |
| Gatsby Charitable Foundation | GAT3395/PR4 | Behruz Bozorg Henrik Jönsson |

The funders had no role in study design, data collection and interpretation, or the decision to submit the work for publication.

### Author contributions

Firas Bou Daher, Conceptualization, Formal analysis, Supervision, Funding acquisition, Investigation, Visualization, Methodology, Writing—original draft, Project administration, Writing—review and editing; Yuanjie Chen, Conceptualization, Formal analysis, Supervision, Investigation, Visualization, Methodology, Writing—original draft, Writing—review and editing; Behruz Bozorg, Conceptualization, Software, Formal analysis, Investigation, Visualization, Methodology, Writing—review and editing; Jack Clough, Conceptualization, Software, Investigation, Visualization, Methodology; Henrik Jönsson, Conceptualization, Software, Supervision, Funding acquisition, Investigation, Methodology, Writing—original draft, Project administration, Writing—review and editing; Siobhan A Braybrook,

Conceptualization, Software, Formal analysis, Supervision, Funding acquisition, Investigation, Visualization, Methodology, Writing—original draft, Project administration, Writing—review and editing

### Author ORCIDs
Firas Bou Daher http://orcid.org/0000-0002-7493-8662
Henrik Jönsson http://orcid.org/0000-0003-2340-588X
Siobhan A Braybrook http://orcid.org/0000-0002-4308-5580

### Decision letter and Author response
Decision letter https://doi.org/10.7554/eLife.38161.022
Author response https://doi.org/10.7554/eLife.38161.023

## Additional files

### Supplementary files
• Supplementary file 1. Relative growth rates (RGR) in length and width for cells by position index along the hypocotyl length, calculated across 6H windows. Propagated standard error of the mean (SEM) values are also provided. Values were calculated from the cell length and width data in *Figure 1*.
DOI: https://doi.org/10.7554/eLife.38161.015

• Transparent reporting form
DOI: https://doi.org/10.7554/eLife.38161.016

### Data availability
All raw data produced and utilized in this study can be downloaded from the Dryad data repository (doi:10.5061/dryad.4s4b3nf). Modeling code can be accessed through the Sainsbury Laboratory's GitLab page (https://gitlab.com/slcu/teamHJ/behruz/3Dhypocotyl; copy archived at https://github.com/elifesciences-publications/3Dhypocotyl).

The following dataset was generated:

| Author(s) | Year | Dataset title | Dataset URL | Database, license, and accessibility information |
|---|---|---|---|---|
| Daher FB, Chen Y, Bozorg B, Clough J, Jönsson H, Braybrook S | 2018 | Data from: Anisotropic growth is achieved through the additive mechanical effect of material anisotropy and elastic asymmetry | http://dx.doi.org/10.5061/dryad.4s4b3nf | Available at Dryad Digital Repository under a CC0 Public Domain Dedication |

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

## Appendix

DOI: https://doi.org/10.7554/eLife.38161.017

Within this work, we utilize both mechanical testing methods and computational modelling that benefits from some further discussion. As such, we have prepared this technical supplement.

## Mechanical modelling of a 3D hypocotyl epidermal cell layer

We developed a 3D finite element method mechanical model to evaluate mechanical signals and growth for cell walls of the epidermal cell layer of a hypocotyl. We used a 3D template and used prisms with 6 walls that are to represent individual cells (*Figure 4*). Each wall is triangulated from its centroid into triangular (planar) elements. The dimensions of the cell walls are proportional to the average values of those seen in experiments, for example *Figure 1*. Individual cells are assembled into a 3D structure representing an epidermal cell layer. A wall in between two cells was divided into two adjacent walls and connected via the corner nodes. In this set up each pair of adjacent walls experiences the same deformation while it can hold individual mechanical properties. The two ends of the template are closed and the tissue is pressurized on the outer surface. The mechanical signals of cells close to each end were excluded from the analysis to avoid artefacts caused by boundary conditions. In order to reduce the boundary effects, vertices at the two ends of the cylinder are constrained to stay in a plane parallel to XY plane while allowed to move freely in the X and Y directions. A mechanical model for planar elements of the cells defines elastic behaviour and material anisotropy (*Bozorg et al., 2014*). The thickness of the walls was included by adjusting their corresponding Young's moduli assuming the material strength is proportional to the amount of material in a unit area.

The relation between stress and strain is described by a St. Venant-Kirchoff strain energy, **W**. For an isotropic material, this is given by

$$W = \frac{\lambda}{2}(trE)^2 + \mu trE^2, \qquad (1)$$

where **E** is the Green-Lagrange strain tensor and $\lambda$ and $\mu$ are the Lame coefficients of the material (*Bozorg et al., 2014*; *Ciarlet, 1993*; *Hamant et al., 2008*; *Sampathkumar et al., 2014*). In the plane stress condition, the Lame constants can be expressed in terms of Young's modulus, *Y*, and Poisson ratio, $\nu$, by

$$\lambda = \frac{Y\nu}{1-\nu^2}, \mu = \frac{Y}{2(1+\nu)}. \qquad (2)$$

In the case of anisotropic material, the following expression is added to the energy

$$\Delta W = \frac{\Delta\lambda}{2}(trE)(a^t E a) + \Delta\mu trE^2 (a^t E^2 a), \qquad (3a)$$

where $a$ and $a^t$ are the anisotropy vector and its transpose (*Bozorg et al., 2014*). The second Piola-Kirchhoff stress tensor, **S**, is calculated from the energy via

$$S = \frac{\partial W}{\partial E}. \qquad (3b)$$

The forces generated by the internal pressure are cancelled by stresses in the individual cell walls. The model assumes mechanical equilibrium where the equilibrium condition is given in the current configuration. For a set of nodes that are connected in the 3D arrangement of cells the equilibrium condition is then given by

$$\sum_{n=1}^{N(g)}\sum_{e=1}^{El(n)}\left(A_e F_e S_e D_{ne} + \frac{1}{3}a_e p n_e\right) = 0. \tag{4}$$

Here, $S_e$ and $F_e$ are the stress and deformation gradient tensor of a planar element $e$, respectively. $O_{ne}$ is the shape vector of element $e$ corresponding to node $n$. $A_e$ and $a_e$ are the resting and current values for the area of the element $e$ respectively, and $n_e$ is the normal vector of the same element pointing outward with respect to the cell volume and $p$ is the pressure. The expression is summed over all of the $El(n)$ elements that are connected via node $n$ and over all of the $N(g)$ nodes in the connected set $g$, that is all nodes representing the same spatial location but connected to different cells.

Growth is modelled using a Lockhart relation (**Lockhart, 1965**), where the strain is used as growth signal (**Bozorg et al., 2016**). The growth is updating the resting shape, $X_0$, by

$$X_0(t+dt) = X_0(t)\left(1 + f_g dt\right) \tag{5}$$

where $dt$ is an artificial time step and $f_g$ is the differential growth tensor. The average strain tensor over all the elements of each cell wall is used as growth signal for those elements. The growth tensor is described by

$$f_g = k_{rate}i\sum R(S_i - S_t)|S_i><S_i| \tag{6}$$

where $S_i$ and $|S_i>$ are the $i^{th}$ principal value and vector of the strain tensor. $R(x)$ is the ramp function ($R(x)=0$ if $x<0$, and $R(x)=x$ otherwise), and $S_t$ is a yield threshold, above which growth will happen.

Strain anisotropy is defined as the measure where, if $S_1$ and $S_2$ are the first and second eigenvalues of the strain in the wall plane

$$a = \frac{S_1 - S_2}{S_1} \tag{7}$$

The mechanical model is simulated until equilibrium (maximal value of LHS of **Equation 4** < eps) using a fifth order Runge-Kutta solver, and growth is simulated by updating the resting configuration at the mechanical equilibrium using a Euler step (**Equation 5**). An in-house developed open source C++ software is used for the simulations (http://dev.thep.lu.se/organism, available upon request). Model output is visualized using Paraview (http://www.parview.org).

**Appendix 1—table 1.** Model parameters for simulations in **Figure 4** and **Figure 4—figure supplement 1**.

| Model parameters for simulations in *Figure 4* and *Figure 4—figure supplement 1* (S4). | Simulation | | | | | |
|---|---|---|---|---|---|---|
| Model parameter | 4A- | 4A+ | 4B1 | 4B2 | S4A- | S4A+ |
| Young's Modulus, inner wall (hoop/trans; $Y_{ref}$) | 1 | 1 | 6 | 1 | 1 | 1 |
| Young's Modulus, inner wall (axial; $Y_{ref}$) | 1 | 1 | 1 | 1 | 1 | 1 |
| Material anisotropy, inner wall | 0 | 0 | 0.83 | 0.83 | 0 | 0 |
| Young's Modulus, outer wall ($Y_{ref}$) | 1.3 | 1.3 | 1.3 | 1.3 | 1.3 | 1.3 |
| Young's Modulus, axial anticlinal wall ($Y_{ref}$) | 1 | 0.1 | 1 | 0.5 | 1 | 0.5 |
| Young's Modulus, transverse anticlinal wall ($Y_{ref}$) | 1 | 10 | 1 | 1 | 1 | 1 |
| Isometric presssure ($P_{ref}$) | 1 | 1 | 1 | 1 | 1 | 1 |
| Extra axial pressure ($P_{ref}$) | 0 | 0 | 0 | 0 | 0.6 | 0.6 |
| Pressure anisotropy | 0 | 0 | 0 | 0 | 0.38 | 0.38 |

$Y_{ref}$ = 50 MPa; $P_{ref}$ = 0.2 MPa

DOI: https://doi.org/10.7554/eLife.38161.018

Cell dimensions were set as: periclinal/anticlinal axial length = 4, periclinal width of outer face = 3, periclinal width of inner face = 2.2, anticlinal depth = 2.2.

## Sensitivity analysis

If $a$ is the strain anisotropy and $p_t$ is the $i^{th}$ parameter, then the $i^{th}$ component of sensitivity was calculated by

$$Sen_i = \frac{\%\Delta a}{\%\Delta p_i} = \frac{100(\Delta a)/a_{old}}{100(\Delta p_i)/p_{iold}}. \tag{8}$$

Since each parameter pi was changed by 10%, we have the case where

$$Sen_i = 10\frac{\Delta a}{a_{old}}|_{(10\% \; increase \; in \; p_i)}$$

Model parameters for sensitivity analysis in *Figure 4—figure supplement 1*: (1) depth of L1 layer (anticlinal walls), (2) transverse anticlinal cell wall density, (3) overall stiffness of inner periclinal walls, (4) overall stiffness of outer periclinal walls, (5) overall stiffness of anticlinal walls, (6) asymmetry of anticlinal walls, (7) pressure. In the analysis of internal wall anisotropy we also included (8) mechanical anisotropy of inner periclinal walls. In the analysis of internal force anisotropy we included (8) anisotropy of internal force, (9) anisotropy of internal force when circumferential E is constant, and (10) anisotropy of internal force when axial E is constant. Parameters were varied 10%, and the relative change in strain anisotropy is reported.

## Technical considerations for Atomic Force Microscopy

### On indentation perpendicular to the direction of growth

As has been described recently (*Cosgrove, 2016*; *Milani et al., 2013*), AFM-based nano-indentation tests mechanical properties by indenting epidermal walls, normal to the wall surface. In many situations, this means the indentation is perpendicular to the direction of growth and as such it is not clear how AFM-based rigidity measurements might relate to growth. What can be said is that thus far, an AFM-based perpendicular-derived elasticity measure (called Indentation Modulus or Apparent Young's Modulus) has correlated well with growth phenotypes observed in all cases reported (*Bozorg et al., 2014*; *Braybrook and Peaucelle, 2013*; *Peaucelle et al., 2011*; *Sassi et al., 2014*). One explanation that might be invoked is that these indentations measure mainly the cell wall matrix which, while heterogeneous, is likely isotropic making its elastic property less sensitive to the direction of loading (*Braybrook and Peaucelle, 2013*). It is also possible that this elasticity is indirectly related to growth and serves as a proxy for other, more dominant, properties such as porosity (*Braybrook et al., 2012*; *Peaucelle et al., 2012*).

### On indentation or retraction analysis

Within this study, we have analysed the indentation portion of each force-deformation curve, as opposed to the retraction. Choosing one over the other is not a straight forward decision. In materials testing, the retraction is often used to calculate the elastic modulus since the indentation portion contains both plastic and elastic components. In biological materials, when the experiments are performed within a linear elastic range, the indentation curve is recommended for analysis as the retraction curve can be influenced by sample geometry and/or sample adhesion (distorting the force-displacement relationship (*Levesque-Tremblay et al., 2015b2015b*). This becomes pertinent in the present study as it relates to an earlier study where retraction curves were analysed for determining the Apparent Young's Modulus (*Peaucelle et al., 2015*). Our analyses indicated that (1) hypocotyl samples were sensitive to adhesion when a 10 nm diameter tip was used and (2) a 10 nm tip was most

relevant when cell walls of <250 nm thickness were being tested (as opposed to the larger 1000 nm tip diameter in (*Peaucelle et al., 2015*). It is most likely that these technical distinctions allowed us to observe an elastic asymmetry in very young cells at 4HPG, where before they were only rarely observed (*Peaucelle et al., 2015*).

## On the effect of spring constant calibration

We calibrate the spring constant of every cantilever used by thermal tuning, the accepted method, and obtain values close to the manufactured specification (~45 N/m). This differs from the universal spring constant adopted in (*Peaucelle et al., 2015*), 1 N/m, irrespective of the supposed manufacturer's range of 42–48 N/m. The specified spring constant would affect the force being applied and thus the deformation obtained between the two studies making it difficult to compare absolute values of IM and possibly contributing to differences in early (pre-24HPG) observations.

## Technical considerations for induction of PME5 or PMEI3 by ethanol

In our growth conditions, after careful selection of germinating seeds at 0HPG (just after endosperm rupture), we were able to achieve 100% of induction (all seedlings were induced). We could visualize this by proxy as the *AlcA::PME5* line also carries an *AlcA::GUS* construct and the *AlcA::PMEI3* line *AlcA::GFP*. The acquisition of 100% induction was critically dependent on the induction vessel and the placement of tubes with inducer: in a standard Petri dish, two 0.5 mL tubes with 200 ul Ethanol were required to be placed on either side of the plate in order to achieve uniform induction. We also observed that if seedlings were induced before endosperm rupture, they were very delayed in growth, possibly due to a rigidification of the endosperm blocking true germination. These observations are perhaps pertinent when comparing our induction results, utilizing the same transgenic lines, to those previously published (*Peaucelle et al., 2015*). We note that in the previous study, a 10 or 20% induction was achieved with *AlcA::PMEI3* or *AlcA::PME5*, respectively (reported as measured in both by GUS staining; note that AlcA::PMEI3 lines do not have a GUS reporter [*Peaucelle et al., 2015*]) by adding 20 ul of ethanol directly to the base of the Petrie dish. Further analysis in the previous study was carried out on seedlings with shorter (PMEI3) or longer (PME) hypocotyls, present in the population to the same frequency of the reported GUS staining (*Peaucelle et al., 2015*). It is possible that growth phenotypes similar to ours were under-represented because they mimicked non-germination (too early induction) or were missed due to low induction penetrance. It is possible that a difference in growth conditions yielded different results. In our study, with 100% induction (GUS or GFP) and 100% phenotype (*PME5*- short, *PMEI3*- long hypocotyls), alongside our immunolocalizations demonstrating that *PME*-induction triggered de-methylation while *PMEI*-induction inhibited it, we feel confident in our interpretation.

Hypocotyl epidermal cell growth has been reported to undergo a switch from isotropic to anisotropic (*Peaucelle et al., 2015*). Our observations demonstrate that hypocotyl epidermal cells are anisotropic in shape from germination and that axial walls consistently elongate faster than transverse walls from germination indicating anisotropic growth. Differences in imaging technique and sample number may have resulted in such differential results: the previous study appears to have used AFM scans to determine cell dimensions with low sample numbers. Since AFM scans are limited in spatial resolution by their acquisition resolution (e.g. a 64 × 64 grid scan on 100 × 100 μm, as reported, would yield a measurement resolution of 1.56 μm). The confocal imaging employed in this study on 20 hypocotyls per time point have provided greater spatial resolution. It is also possible that different growth conditions may account for the different results.

## Technical considerations for cell wall immunolocalizations

### On the interpretation of cell-wall immunolocalization

In the present study, we have chosen to utilize two complimentary cell-wall homogalacturonan (HG) anti-bodies: LM19 which has a preference and strong binding for de-methylated HG, and LM20 which requires methyl-esters for recognition of HG and does not bind to de-methylated HG (*Verhertbruggen et al., 2009*). De-methylated pectin has the potential to bind calcium and increase pectin gel rigidity. In our experiments, we see co-incidence of high LM19 (and low LM20) with higher cell wall rigidity. It is our opinion that complimentary anti-bodies provide more information than either one singly. To confirm our hypothesis, that PME activity in the hypocotyl led to de-methylation and rigidity increase, we performed immunolocalization with the 2F4 antibody (recognizes calcium cross-linked HG) on 24HPG hypocotyls and demonstrated a pattern in line with our prediction (slow-growing cells had higher 2F4 signal). All of our immunolocalizations were performed on LR White embedded material, as in (*Verhertbruggen et al., 2009*); we have concerns that since 2F4 immunolocalizations require calcium in their buffer, in more malleable wax embedding one might observe the HG that can bind to calcium in vitro in combination with that which was bound *in planta*. It is perhaps for this reason that previously published immunolocalizations differ from our own (*Peaucelle et al., 2015*) although it is hard to assess the published images for cell type, section plane, or location within the hypocotyl.

Lastly, as mentioned in the text the antibodies available for cell wall epitopes specifically recognize specific cell wall epitopes. As such, while they provide an idea of pectin biochemistry they do not describe all pectin within the cell walls being probed. Methods such as FT-IR spectroscopy (*Mouille et al., 2003*) provide a more thorough picture of pectin biochemistry but they lose the spatial context which immunolocalisations afford. As developmental biologist, we prefer the spatial context but must recognize the limitation of such mono-clonal antibodies. None the less, we present a striking correlation between growth, wall elasticity, and LM19/2F4 and LM20 signals in the elongating dark-grown hypocotyl. Our results are also firmly in line with in vitro data on pectin gel methylation and mechanical properties.

## Technical considerations for imaging and measuring MT alignment

### On choosing a marker

In this manuscript we present data from plants expressing 35S::GFP-MAP4; this line is notorious (within the community) for having some phenotypic deviations from the non-transgenic. In the case of the hypocotyl, plants will often have swollen cells indicating a disruption of some normal MT function. We specifically select for individuals in our seeds stocks which do not show this phenotype. In order to make sure we aren't looking at a transgenic artifact, all of our MAP4 data was backed-up by analyses of two other MT markers: GFP-TUA6 and GFP-EB1. MAP4 binds to microtubules along their lengths, EB1 binds to their growing ends, and TUA6 is a tubulin that is incorporated into microtubules themselves. Lastly, we did some analyses of CESA3::CESA3:GFP expressing hypocotyls as well; this was done in order to confirm that MT pattern and CESA pattern matched in our conditions, but also because it is technically impossible the perform FE-SEM on seedlings as young as ours and so CESA3 tracks were the closest we could get to visualizing cellulose. As we note in the text, cellulose microfibrils could move and alter orientation once in the cell wall and at this time we have no way to observe whether such re-arrangements might occur.

## On strange expression phenomena in the very young hypocotyl

In our imaging of GFP-MAP4 we were surprised that we could not visualize MTs at the inner epidermal wall of young (<24 HPG) seedlings. To our knowledge, our experiments are the first to attempt imaging this early of MTs in basal hypocotyl cells and so this has never been reported before. We confirmed this with GFP-TUA6 as well: while signal was strong at the outer epidermal face, it diminished with depth but re-emerged once the cortical cells were encountered. The signal from cortical cells was strong and so we did not lose signal strength towards the bottom of the epidermal cells. The cells were obviously cortical as they are wider than the epidermal cells above them. While this may be an artefact of transgenic localization, it is also possible that MTs are preferentially localized to the outer epidermal face during early hypocotyl growth in order to reinforce the outer wall.

## On choosing an analysis method

There are several methods to choose from when analyzing MT angle/orientation. Two of the most frequently used are methods which measure MT angle distribution (e.g. MicroFilament Analyzer, used here) and FibrilTool (*Boudaoud et al., 2014*). In our conditions, which included very different imaging planes (outer versus inner walls) and therefore different amounts of noise as well as different cell sizes, we found FibrilTool to be less useful. Testing its performance with synthetic images (drawn lines, etc.) we found it to be sensitive to background noise and fiber length. As such, we report only microtubule angle in this manuscript. FibrilTool is likely very useful when imaging conditions and cell sizes are consistent. Lastly, we note in the text that no one really knows what value of angles or Fibril Tool anisotropy degree actually confer anisotropic properties to cell walls. This means that there is still a large conceptual leap between an angle distribution, or anisotropy value, and the actual physical growth process. Further complications arise when one considers that in many tissues and cells, MT orientation is not always the same as cellulose fiber orientation.

