## [Decision Letter]

Thank you for submitting your article "Anisotropic growth is achieved through the additive mechanical effect of material anisotropy and elastic asymmetry" for consideration by *eLife*. Your article has been reviewed by two peer reviewers, including Dominique Bergmann as the Reviewing Editor and Reviewer #1, and the evaluation has been overseen by Christian Hardtke as the Senior Editor. The following individual involved in review of your submission has agreed to reveal his identity: Geoffrey O Wasteneys (Reviewer #2).

The reviewers were quite positive about this work and look forward to seeing an updated version soon!

Summary:

Plants grow in stereotyped ways in response to developmental and environmental cues. Such is the case for hypocotyl growth during seedling emergence. This relatively simple system has become the model of choice for examining cellular growth in the absence of cell division. This paper looks at some of the mechanical aspects of growth, a phenomenon that has, in the past, been described in rather extreme views – either something due to cellulose orientation or due to pectin-mediated stiffness.

The strengths of this work are that it takes a multi-faceted approach to describing hypocotyl growth by observation of dynamics, by labeling of cell wall epitopes, and by probing of stiffness. The strategies are comprehensive and economical. The experimental evidence is informed by the modelling approach, which has nicely dissected the different parameters. The use of inducible activity of pectin methylesterification activity, in combination with atomic force microscopy, backed up with epitope-specific immunofluorescence assays is elegant. The authors have carried out meaningful experiments with appropriate controls, and have identified any technical limitations to preclude the need for further analysis at this time. Another strength is that this work is open about ambiguous and conflicting data and addresses the contradictions of the data here and with reference to other papers head on.

Essential revisions:

The reviewers agreed that there were no additional experiments needed. One of us thought the point that multicellular organs have "collective" behaviors or properties that don't require each cell to have identical material properties was really interesting. This was one place where a bit of extra modeling on how an organ could reach a particular form via changes to properties in a few cells would be a powerful addition and would cement this article as the place this concept was laid out first/best.

---

## [Author Response]

Essential revisions:The reviewers agreed that there were no additional experiments needed. One of us thought the point that multicellular organs have "collective" behaviors or properties that don't require each cell to have identical material properties was really interesting. This was one place where a bit of extra modeling on how an organ could reach a particular form via changes to properties in a few cells would be a powerful addition and would cement this article as the place this concept was laid out first/best.

We agree that modeling showing how a few cells could dictate a change in form would be useful, so much so that we are currently working on this exact question in two contexts: the hypocotyl apical hook and circumnutating hypocotyls. However, both of these projects are at least a year out from being ready for inclusion in publication. An additional point is that the modeler for this manuscript, Bozorg, has moved on to another position making additional modeling here a larger task. We look forward to publishing these examples as future work. We also note that something along this line has been published by the group of Jan Traas, in organogenesis at the shoot meristem, where changing the mechanics of a group of cells was sufficient to generate a bulge (http://journals.plos.org/ploscompbiol/article?id=10.1371/journal.pcbi.1003950).